



**Understanding Spatial Variations in Earthquake Vulnerabilities of**
**Residential Neighborhoods of Mymensingh City, Bangladesh: An AHP-GIS**
**Integrated Index-based Approach**
Md. Shaharier Alam[1], Shamim Mahabubul Haque[2]
[1]Assistant Urban Planner, Asian Disaster Preparedness Center, Email address: shaharier3@gmail.com
[2]Professor, Urban and Rural Planning Discipline, Khulna University, Khulna-9208, Bangladesh, Email address:
shamimhaque67@yahoo.com
**Abstract:** Mymensingh city is highly earthquake vulnerable due to its geological setting, existence of three
faults, viz., Dauki Fault, Madhupur Blind Fault and Sylhet-Assam Fault in its close vicinity, and liquefaction
susceptible soil type. Recently an attempt has been made to assess earthquake risk of the city by Comprehensive
Disaster Management Programme II, of Government of Bangladesh using FEMA developed HAZUS tool which
requires usage of enormous resources and expertise. Poorly resourced city planning authorities of developing
countries are seldom equipped with such financial and human resources, and as a result, the inclusion of
earthquake risk analysis, more specifically, information regarding spatial variations of earthquake risk is very
often found missing in their physical planning exercises. This paper aims to assess the spatial variation of
earthquake vulnerability of residential neighbourhoods of Mymensingh city, employing an index-based low cost
approach which could provide a reasonably accurate result with minimum resource and expertise requirements.
Analytical Hierarchy Process and Weighted Linear Combination are combined with a Geographical Information
System to prepare a composite index considering 23 different parameters, stemming from geological, structural,
socio-economic and systematic dimensions of earthquake vulnerability. The findings of the reseach show that
out of 241 residential neighbourhoods of Mymensingh city, 51 are observed to be highly vulnerable, while, 123
and 67 are medium and low vulnerable respectively. Besides, the spatial distribution of earthquake vulnerable
neighbouhoods in Mymensingh City, observed in the current study has also been compared with spatial
distributions observed in two similar previous studies and observed found to be reasonably close. This justifies
the validity of the current low cost approach for wider application in cities of resource starved developing
countries.
**Keywords-** Earthquake vulnerability, Index, AHP, GIS, WLC, City planning and development

## 1. Introduction

Bangladesh, the largest delta of the world, is prone to numerous natural catastrophes due to its geographical
location, and remarked as the 5[th] most disaster risk zone by Asia Pasific Disaster Report 2017(ESCAP,2017).
Understanding the complexity of vulnerability caused by various natural disasters is the most challenging task of
disaster risk reduction of an area (Alam, Chakraborty and Islam,2019). Earthquake is one of the most lethal
disasters, specially in the contex of Bangladesh as the country has been shaken up by more than 250 earthquakes
since her independence (Zaman et al., 2018). Tectonically, the country lies at the junction of three tectonic
plates - the Indian Plate, the Eurasian Plate, and the Burmese micro-plate, which puts the country in one of the
most tectonically active regions of the world (Al Zaman and Monira,2017). A recent GPS measurement of plate
motions in Bangladesh combined with measurements from Myanmar and northeast India, reveal 13–17mm/yr of
plate convergence on an active, shallowly dipping and locked megathrust fault underneath of Bangladesh which
could unleash a 9-magnitude earthquake at any time and kill ten million people (Steckler et al. 2016). The city
of Mymensingh is located in zone IV (seismic coefficient 0.36g) of seismic macro-zonation map of Bangladesh
and is demarcated as one of the most earthquake-vulnerable cities of the country (BNBC, 2015). The city is
seismically vulnerable due to its proximity to three major faults viz. Madhupur Blind Fault, Dauki Fault, and
Sylhet-Assam Fault. Besides, liquefaction susceptible soil type covers almost 90 percent of the total area of the
city which adds a new dimension to the earthquake vulnerability of the city. Not only the geological factors
lying beneath the earth's surface but also factors lying above the earth surface, such as structural, socio-
economic and systematic factors are making Mymensingh City vulnerable to earthquake and puts lives and
assets of its citizen at risk. Mymensingh, being one of the oldest municipalities of Bangladesh, is vulnerable due
to thousands of old dilapidated buildings that are at particular risk of collapse. Besides, substantial variations in
socio-economic conditions among residential neighbourhoods are also observed across the city. Considering its
increasing administrative importance, and economic potentials, the city has recently been elevated to the status



of the 8th divisional city of Bangladesh (Alam and Haque, 2017). The city is expected to house a population of
3 million by the end of the year 2021 which would also open up possibilities of mass migration, haphazard
development, and unplanned future expansions.
Residential neighbourhoods of the cities are generally highly vulnerable to earthquake due to their high spatial
concentration of life and assets. Nwe and Tun (2016) examined the seismic vulnerability of Mandalay city based
on land use condition and observed that residential land use type is the third seismically vulnerable land use type
of a city after mixed-use (resident with a store) and commercial land use types. As an old and historic city of
Bangladesh, the buildings in the residential neighbourhoods are old in Mymensingh, and substantial
socioeconomic disparities among the neighbourhoods are observed. Therefore, given historical and increasing
administrative importance of the city, it is crucial to assess all dimensions of earthquake vulnerabilities and their
spatial distribution across the city to prioritise earthquake risk reduction strategies for the city.

## 62  2.  Literature Review

### 63  2.1. Rationale

Earthquake vulnerability can be precisely assessed using HAZUS, a Geographic Information System (GIS)
based multi-hazard risk assessment tool developed by the Federal Emergency Management Agency (FEMA) of
the United States of America. The HAZUS methodology has capabilities to assess the spatial variations of,
among others, earthquake, flood, hurricane risks through following several steps such as study region definition,
hazard characterisation, and damage and loss estimation. But HAZUS cannot be readily used in other countries
due to unavailability of boundary characterization function outside the USA. Therefore, it is opined that HAZUS
can provide only a starting point for the development of a disaster risk assessment tool which could be used in
Bangladesh considering user requirements and data availability (Sarker, Ansary, Rahman & Safiullah 2009).
Another significant complexity of using HAZUS is the development of fragility function which requires a huge
amount of resources, high-level of expertise and an enormous amount of data. Developing countries like
Bangladesh are hardly equipped with this type of resource, data, and expertise. This paper primarily focuses on
developing less resource, data and expertizes requiring methodology to assess earthquake vulnerabilities at
neighborhood scale and observe their spatial distribution across the city. The developed methodology is applied
to assess spatial variations in earthquake vulnerabilities of residential neighbourhoods of Mymensingh City
which yielded a reasonably accurate result and ushered in the possibility of its use in planning efforts of cities
having poorly resourced planning agencies in the developing counties.

### 80  1.2. Dimensions of Earthquake Vulnerability Assessment

Overall earthquake vulnerability of a neighbourhood largely depends on its structural, geological, socio-
economic and systematic components. Excluding any one of these components may have severe implications in
devising appropriate risk reduction strategies at the city level. Researchers all over the world are working on the
evaluation of earthquake vulnerability using different methods and dimensions. Unfortunately, most of the
research work on earthquake vulnerability is focused on structural component and hardly consider other
dimensions of vulnerability. Sarvar, Amini, and Laleh-Poor (2011) assessed the earthquake risk of Tehran using
a hybrid methodology which only considered structural dimensions of the area. Lantada, Pujades, Barbat (2004)
also evaluated the seismic risk of Barcelona using the vulnerability index method and capacity spectrum-based
method which had been structural vulnerability biased and excluded socio-economic dimension of the area.
Researchers such as Nath et al. (2015), Ishita and Khandakar (2010), Barbat et al. (2010), Sarris et al. (2010)
also attempted to measure seismic vulnerability at different spatial scale but only considered the structural or
geological dimension of vulnerability and excluded socio-economic dimension of an area. On the contrary,
researchers including Armas and Gravis (2013); Martins, de Silva and Cabral (2012); Walker et al. (2014),
Shirley, Boruff and Cutter (2012), Pelling (2012) in their researches highly focused on the social dimension of
vulnerability of natural hazard and undervalued the other dimensions. At city scale, especially in case of cities of
developing nations, it is essential to combine all dimensions of earthquake vulnerability to get a complete
picture of overall vulnerability situation and its spatial implications to devise appropriate development control
mechanism and resource targeting. Moreover, the studies mentioned above are not land use specific which is a
major short coming for undertaking any city level land use micro-zonation, since vulnerability significantly





varies with the pattern of land use also. This study endeavours to assess the land use specific earthquake
vulnerability of Mymensingh City combining all dimensions of vulnerability including structural, geological,
socio- economic and systematic dimensions.
**1.3. Methods of Earthquake Vulnerability Assessment**
While assessing overall vulnerability, it is always difficult to find an appropriate methodology since most of the
contemporarily developed methods cannot integrate revealed and stated preference data at a time. The data type
varies along with the vulnerability dimensions considered. Most of the structural, systematic or geological data
of earthquake vulnerability are revealed preference whereas socio-economic data are both stated and revealed
preference data. VahidiFard et al. (2017), Bessason and Bjarnason (2016) analysed the seismic risk of an area
using time series data and damage data of previous high magnitude earthquake. Unavailability of data restricts
the use of this method in developing nations like Bangladesh.  Whitman et al. (1973), Braga et al. (1982),
Lantada et al. (2010) used damage probability matrix to evaluate the earthquake risk which only considered the
structural vulnerability and requires post-earthquake building damage statistics. Freeman et al. (1975) used the
Capacity Spectrum Method (CSM) to evaluate probable seismic vulnerability by developing a capacity curve
and demand curve which is a very complex methodology. Federal Emergency Management Agency (2015) has
developed a method of rapid visual screening (RVS) to assess the seismic vulnerability which does not require
historical or damage data of the previous earthquake but requires every detail of building stock which is very
time and resource consuming. There are several other methods such as Non-linear Dynamic Analysis (Fajfar,
2000), Vulnerability Index Method (Lantada, 2010), Failure Mechanism Identification and Vulnerability
Evaluation (FaMIVE) method (Formisano, Mazzolani &Indirli, 2010), etc. available for seismic damage
evaluation. But all these methods are complicated, time-consuming, require high-level expertise and data
support, and most importantly all of them are structural vulnerability component biased. Methods of analysis
deployed in many of the reported vulnerability analysis are very complex requiring specific skill and expertise
which may not be inplace for many developing countries.
Moreover, most of the reported works on earthquake vulnerability are not land use specific.  Therefore, a simple
but efficient methodology which can incorporate all the issues mentioned above of earthquake vulnerability
assessment is needed for the use in the planning process of cities of developing nations. Multi-criteria decision
making (MCDM) is the simplest and efficient methods used by researchers to integrate all dimensions of
vulnerability as it can solve complex decision-making covering a wide range of choices and prioritising of
decision-making alternatives (Rezaie and Panahi,2015). Analytical Hierarchy Process is the most renowned and
comprehensive MCDM procedure which can integrate both stated and revealed preference data simultaneously
and hierarchically solves complex decision-making issues by developing a pairwise comparison matrix.
Weighted Linear Combination (WLC), another simple additive MCDM method, generally used with AHP
method to get a composite score by multiplying the weight of the criteria and sub-parameters.
In this paper, spatial variations of earthquake vulnerabilities of the residential neighbourhoods of Mymensingh
City have been assessed by integrating an index-based approach and GIS analysis. Analytical hierarchy process
(AHP) and Weighted Linear Combination (WLC) methods have been used to develop an index combining four
dimensions of vulnerability. At first, four different indices, viz., structural vulnerability index, socio-economic
vulnerability index, geological vulnerability index and systematic vulnerability index are developed using expert
opinions based AHP method. Then a composite index is developed using WLC method combining all four
indices based on expert opinions and spatial variation of earthquake vulnerability among residential
neighbourhoods of Mymensingh are analysed and visually presented in the map using GIS technology. Finally,
the result obtained from this study has been compared with the previously reported assessments of the same
study area done by CDMP-II and Sarker et al. (2009) using Cohen kappa statistics and confusion matrix. All
results are found to be reasonably close which justifies the validity of the current approach.






## 3. Methodology

### 3.1. Study Area

The city of Mymensingh is the oldest municipality and latest administrative division of Bangladesh, which is located in the northern part of the country (24°45' N latitude and 90°23'E longitude) on the bank of old Brahmaputra River. The city established in 1787 and became a municipality in 1869, has an area of 2.73 sqkm. has a population of 258,040 (Male-132,123, Female-125,917) and has a population growth rate of 1.82% (BBS, 2011). The city experienced earthquakes in the past including 1762 earthquake (7.5 Mw) originated from the Madhupur tract in which the course of the river Brahmaputra changed dramatically and the Great Indian earthquake of 1897 (8.7 Magnitude) in which the whole Mymensingh City was collapsed (CDMP, 2014). There are 21 administrative wards, and 241 residential neighbourhoods in Mymensingh city

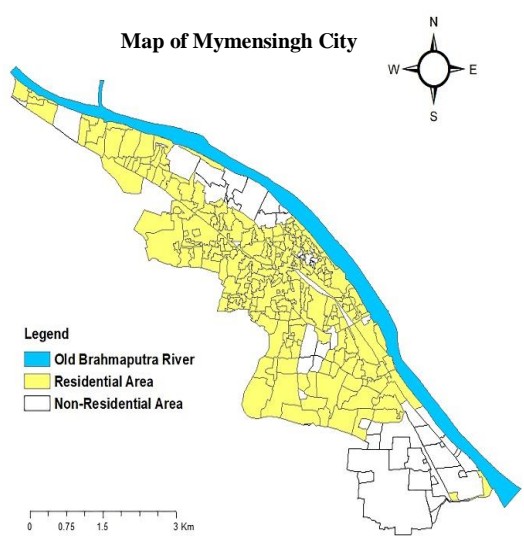

**Fig. 1:** Residential neighborhoods of Mymensingh city

(**Fig. 1**), delineated based on metal space mapping during the preparation of the Mymensingh Strategic Development Plan (MSDP) sponsored by the Comprehensive Disaster Management Program (Phase-II) of the Government of Bangladesh.

### 3.2. Selection of Parameters of Earthquake Vulnerability Assessment

In this study, 23 influential earthquake vulnerability parameters have been selected based on diligent literature review, expert opinion and by analysing available data, under four vulnerability dimensions, viz., geological, structural, socio-economic and systematic vulnerability.

### 3.2.1. Geological earthquake vulnerability parameters

Geological parameter refers to the factors related to the earth that affects the earthquake vulnerability of an area. The geological parameters considered in this study are shown in **Table 1**.

**Table 1** Geological Earthquake Vulnerability Parameters

| Parameter | Vulnerability Level | | | Supporting Literature |
|---|---|---|---|---|
| | Low | Medium | High | |
| **Soil Type** | Hard Soil | Stiff Soil | Soft Soil | Isihita and Khandakar2010; Sarvar, Amini, and Laleh-Poor2011; Vicente et al.2010; Maddox,2015; |
| **Peak Ground Acceleration** | 0.346485 - 0.369287 | 0.369288 - 0.392051 | 0.392052 - 0.410747 | Rezaie and Panahi2015; Habibi et al.2014; Peek-Asa et al.2003; Moradi, Delavar and Moshiri,2014 |
| **Shear Wave Velocity** | More than 360 m/s | 180m/s to 360 m/s | less than 180m/s | Capilleri, Maugeri and Raciti, 2010; Martin and Diehl,2004 |

This study excludes some other most critical geological parameters including earth slope, depth of water table, etc. due to data unavailability or rare existence in Mymensingh city.

### 3.2.2. Systematic Earthquake Vulnerability Parameters

One of the influential earthquake response issues in cities is the accessibility of residential neighbourhoods to different infrastructure and service facilities such as medical care facilities, open spaces, road networks, fire service, emergency shelter, etc. (Raizee and Panahi,2015). These physical accesses to critical facilities are referred as systematic vulnerability, focusing on rapid post seismic building usability assessment, number, and quality of temporary shelters, accessibility to work sites and services from temporary shelters and vulnerability of strategic public facilities (Atun and Menoni, 2014). Parameters considered for assessing systematic earthquake vulnerability are shown in **Table 2**.




**Table 2** Systematic Earthquake Vulnerability Parameters

| Parameter | Vulnerability Level | | | Supporting Literature |
|---|---|---|---|---|
| | Low | Medium | High | |
| **Distance to hospital** | <500m | 500m to 1km | >1km | Daneshvar, Rezayi, and Khosravi2013; Bac–Bronowicz and Maita, 2001 |
| **Distance to Fire Service** | <1km | 1km to 1km | >2km | Armas,2012; Scawthorn, Eidinger& Schiff, 2005 |
| **Distance to Emergency center** | <500m | 500m to 1km | >1km | Rezaie and Panahi,2015;Atun and Menoni, 2014; Alam and Haque 2018 |
| **Distance to Evacuation Route** | <500m | 500m to 1km | >1km | Bac–Bronowicz and Maita, 2001, Meshkini, Habibi and Alizadeh, 2013 |


### 3.2.3. Structural Earthquake Vulnerability parameters
Structural earthquake vulnerability parameter refers to the factors that relate to the built up environment such as
buildings, bridge, road, etc. Structural parameters have a great influence on earthquake vulnerability and
damage potential of a neighbourhood. In this study, eight most influential structural parameters are considered
to assess the earthquake vulnerability of Mymensingh city which is shown in **Table 3**.
**Table 3** Structural Earthquake Vulnerability Parameters

| Parameter | Vulnerability Level | | | Supporting Literature |
|---|---|---|---|---|
| | Low | Medium | High | |
| **% of poor building** | < 25% | 25 to 50% | > 50% | Moradi, Delavar and Moshiri(2014), Ghajari et al.( 2017), Güzey et al.(2013), Ebrahimian-Ghajari et al.( 2015) |
| **% of BFL Building (masonry building with flexible roof) building** | < 25% | 25 to 50% | > 50% | Isihita and Khandakar(2010), Rahman, Ansary and Islam (2015) |
| **Average Building Storey** | 1 Stroey | 2 Storey | ≥3 story | Sarris et al.,(2009), Vicente et al., (2010), Nath et al., (2015), Isihita and Khandakar(2010) |
| **Average Road Width(ft.)** | >16ft | 8ft to 16ft | <8ft | Isihita and Khandakar(2010) ,Ghajari et al., (2017) |
| **Building Density/acre** | <10 building | 10 to 15 building | >15 building | Zebardast (2012), Armaş (2012) , Martins, e Silva and Cabral,(2012) |
| **Irregular Shape Building (%)** | <10 % | 10 to 15 % | >15 % | Güzey et al., (2013), Ferreira et al.,(2013), Maio et al.,(2015) |
| **Pounding Possibility (%)** | <10 % | 10 to 15 % | >15 % | Jeng and Tzeng, (2000), Ahmed, Jahan and Alam,(2014) |
| **Heavy Overhanging (%)** | <10 % | 10 to 15 % | >15 % | Ahmed, Jahan and Alam, (2014), Güzey et al.(2013) |

Some other most crucial structural vulnerability parameters such as- soft storey, short column, the age of a
building, lateral stiffness, existence of the shear wall, etc. are excluded from this study due to data unavailability
or rare existence in residential neighbourhoods of Mymensingh city.
### 3.2.4. Socio-economic Earthquake Vulnerability Parameters
Unfortunately, during recent years, earthquake experts have not paid enough attention to socio-economic
dimensions of earthquake vulnerability, and therefore only a handful of studies have been conducted in this
regard. The socio-economic vulnerability parameters that are considered in this study are mentioned in **Table 4**.
**Table 4** Socio-Economic Earthquake Vulnerability Parameters

| Parameter | Vulnerability Level | | | Supporting Literature |
|---|---|---|---|---|
| | Low | Medium | High | |
| **Percentage of child Population(<5 yr)** | <5% | 5 % to 10% | >10% | Zebardast,(2012), Rahman, Ansary and Islam,(2015) |
| **Percentage of Elderly population(65+yr)** | <2.4% | 2.4% to 4.8% | >4.8% | Zebardast, (2012), Armaş and Gavriş,(2013) |
| **Women population (%)** | <25% | 25% to 50% | >50% | Armaş et al.,(2017), Schmidtlein et al.,(2011) |




| Literacy Rate | >70% | 35% to 70% | <35% | Güzey et al., (2013), Islam, Swapan and Haque, (2013); Fatemi et al. 2017 |
|---|---|---|---|---|
| **Average Household income** | >16475BDT | 8238 BDT to 16475 BDT | <8238BDT | Armaş and Gavriş,(2013), Duzgun et al.,(2011); Rahman, Ansary and Islam,(2015) |
| **Population Density/acre** | <100 person/acre | 100 to 150 person/acre | >150 person/acr e | Barbat et al.,(2010), Nath et al.,(2015), Armaş and Gavriş,(2013) |
| **Average Household size** | <2.21 | 2.21 to 4.41 | >4.41 | Schmidtlein et al.,(2011), Armaş,(2012),Güzey et al.,(2013) |
| **Religion** | Islam | Sanatan | Others | Atun and Menoni,(2014), de Ruiter et al.,(2017) |
| **Economically dependent population (%)** | <25% | 25% to 50% | >50% | Kalaycioglu, 2006; Armaş et al.,(2017), Moradi, Delavar and Moshiri,(2014), Martins, e Silva and Cabral,(2012), Walker et al., (2014) |


**3.3. Method**
**3.3.1. Analytical Hierarchy Process**
In this study, the Analytical Hierarchical Process (AHP) is used to develop indices to measure spatial variations
of earthquake vulnerabilities of the residential neighbourhoods of Mymensingh city. AHP is a widely used
multi-criteria decision-making method (MCDM) of vulnerability assessment due to its simplicity and rationality
(Alam and Mondal,2018) which considers both qualitative and quantitative parameters to develop a hierarchical
solution in decision making among various alternatives and its sub-category. Analytical Hierarchical Process
(AHP) uses the opinions of experts to weight vulnerability parameters and sub-parameters, and as a result,
transparency and consideration of local socio-economic condition, special conditions of the study area are
ensured that global indices cannot consider (Füssel, 2010). Three major steps are followed by the AHP model in
assessing earthquake vulnerability which are;
**First step**- The first step is the generation of binary comparison matrices on a scale of 1–9 developed by Saaty,
(1980) in which 1 indicating that the two parameters are equally important, and, 9 implying that one parameter
is more important than another. The scale of importance is shown in **Table 5**.
**Table 5:** Magnitude of importance for pairwise comparison (Saaty, 1980)

| Decreasing Relative Intensity of Importance | Equally Important | Increasing Relative Intensity of Importance |
|---|---|---|
| 1/9  1/8  1/7  1/6  1/5  1/4  1/3  1/2 | 1 | 2  3  4  5  6  7  8  9 |

**Second step-** In the second step, weights of different parameters are calculated from the row-multiplied value
(RMV), in unnormalized and normalised values using the following eq-1 and 2.
$$\text{Unnormalized value, } m_i = \sqrt[n]{RMV} \qquad (1)$$
$$\text{Normalized value} = \frac{mi}{\sum_{i=1}^{n} mi} \qquad (2)$$
Here $m_i$ refers to the unnormalized value of the $i^{th}$ parameter and n represents the total influential parameters.
**Third step-** The most important issue in weighting the factors is the consistency between judgments and
weights which is done in the $3^{rd}$ step. The consistency is measured using consistency index and consistency ratio
using eq-3 &4. If the consistency ratio is greater than 0.1, the matrix has inconsistency, and pairwise comparison
must be reperformed between indicators and sub-indicators.
$$\text{Consistency index, CI} = \frac{L-n}{n-1} \qquad (3)$$
$$\text{Consistency ratio, CR} = \frac{CI}{RI} \qquad (4)$$



L represents the Eigenvalue of the pairwise comparison matrix, and RI is the random inconsistency index, which
has some developed value and depends on the number of vulnerability assessment parameters (N). The
variations of RI value for different parametersare shown in **Table 6**.
**Table 6:** Random inconsistency indices (RI) for n = 1, 2, . . ., 12. (Saaty, 1980)

| N | 1 | 2 | 3 | 4 | 5 | 6 | 7 | 8 | 9 | 10 | 11 | 12 |
|---|---|---|---|---|---|---|---|---|---|----|----|----|
| RI | 0 | 0 | 0.58 | 0.90 | 1.12 | 1.24 | 1.32 | 1.41 | 1.45 | 1.49 | 1.52 | 1.54 |

**3.3.2. Weighted Linear Combination**
WLC technique is an additive weighting method in which a weight is assigned to each factor at the initial stage.
The weight of vulnerability parameters determined by using AHP method based on expert opinions is used with
their corresponding individual standardised criteria as input for the WLC aggregation method. In the final step
in developing the earthquake vulnerability map, all the weighted layers are combined using a weighted overlay
technique in the ArcGIS platform. The final vulnerability score is determined according to the linear addition of
given weight to all parameters and their sub-categories (according to Eq. 5).
$$W=\sum_{j=1}^{n} W_j w_{ij} \tag{5}$$
Here W shows the index value of each neighbourhood in vulnerability map, $W_j$ shows the normalised weight of
each parameter, $w_{ij}$ is the weight of $i_{th}$ sub-category related to the $j_{th}$ parameter and n denotes the total number of
influential parameters.
In this study, comparison matrices of 23 earthquake vulnerability parameters (3 Geological, 8 Structural, 8
Socio-economic and 4 Systematic vulnerability parameters) are developed based on judgments of 3 experts.
Then, to aggregate opinions into one matrix, geometric means of the expert's opinion are calculated (Shown in
**Table 7,Table 8,Table 9** and **Table 10**). The aggregated comparison matrix of earthquake vulnerability
assessment used in this study is shown in **Table 11**.
**Table 7:** Pairwise comparison matrix, weight and consistency ratio of Geological earthquake vulnerability
parameters based on the expert's opinion

| Geological Parameters | PGA | Soil Type | SWV | Weight |
|---|---|---|---|---|
| Peak Ground Acceleration (PGA) | 1 | 0.63 | 1.59 | .318 |
| Soil type | 1.59 | 1 | 2 | .466 |
| Shear Wave Velocity (SWV) | 0.63 | .5 | 1 | .216 |
| (Consistency Ratio=0.003, Random inconsistency=0.58) | | | | |

**Table 8:** Pairwise comparison matrix, weight and consistency ratio of Systematic earthquake vulnerability
parameters based on the expert's opinion

| Systematic Parameters | Hospital | Fire service | Shelter | Route | Weight |
|---|---|---|---|---|---|
| Distance to hospital | 1 | 0.55 | 1.82 | 1.26 | 0.253 |
| Distance to fire service | 1.82 | 1 | 1.82 | 1.82 | 0.374 |
| Distance to emergency shelter | 0.55 | 0.55 | 1 | 0.69 | 0.162 |
| Distance to Evacuation route | 0.79 | 0.55 | 1.44 | 1 | 0.211 |
| (Consistency Ratio=0.014, random inconsistency=0.9) | | | | | |

**Table 9:** Pairwise comparison matrix, weight and consistency ratio of structural earthquake vulnerability
parameters based on the expert's opinion

| Structural Parameters | 1 | 2 | 3 | 4 | 5 | 6 | 7 | 8 | Weight |
|---|---|---|---|---|---|---|---|---|---|
| 1. Building Storey | 1 | 0.29 | 0.55 | 0.29 | 0.69 | 0.69 | 0.63 | 1.82 | 0.074 |
| 2. Poor conditioned building | 3.44 | 1 | 1.44 | 0.69 | 1.14 | 1.25 | 0.87 | 1.25 | 0.143 |
| 3. BFL building | 1.81 | 0.69 | 1 | 0.31 | 0.48 | 0.63 | 0.5 | 1.82 | 0.088 |
| 4. Pounding | 3.44 | 1.44 | 3.22 | 1 | 1.59 | 2.62 | 1 | 2.28 | 0.213 |




| 5. | Irregular shaped building | 1.45 | 0.88 | 2.08 | 0.63 | 1 | 1 | 0.55 | 1.26 | 0.116 |
|---|---|---|---|---|---|---|---|---|---|---|
| 6. | Overhanging | 1.45 | 0.8 | 1.59 | 0.38 | 1 | 1 | 0.55 | 3.12 | 0.118 |
| 7. | Road width | 1.59 | 1.15 | 2 | 1 | 1.82 | 1.82 | 1 | 2.88 | 0.178 |
| 8. | Building Density | 0.55 | 0.8 | 0.55 | 0.44 | 0.79 | 0.32 | 0.35 | 1 | 0.068 |
| (Consistency Ratio=0.034, Radom Inconsistency=1.41) | | | | | | | | | | |

**Table 10:** Pairwise comparison matrix, weight and consistency ratio of Socio-economic earthquake vulnerability parameters based on the expert's opinion

| Socio-economic parameters | 1 | 2 | 3 | 4 | 5 | 6 | 7 | 8 | 9 | Weight |
|---|---|---|---|---|---|---|---|---|---|---|
| 1. Household income | 1 | 2.62 | 1.26 | 0.19 | 0.19 | 1.26 | 0.32 | 1.26 | 3.56 | 0.072 |
| 2. Household size | 0.38 | 1 | 0.33 | 0.18 | 0.18 | 0.44 | 0.26 | 0.38 | 1.26 | 0.034 |
| 3. Population density | 0.79 | 3.00 | 1 | 0.28 | 0.28 | 1.26 | 0.40 | 1.26 | 3.56 | 0.077 |
| 4. Elderly population | 5.19 | 5.59 | 3.56 | 1 | 1.00 | 3.00 | 2.00 | 3.56 | 5.59 | 0.258 |
| 5. Child Population | 5.19 | 5.59 | 3.56 | 1.00 | 1 | 3.00 | 2.00 | 3.30 | 5.19 | 0.255 |
| 6. Dependent population | 0.79 | 2.29 | 0.79 | 0.33 | 0.33 | 1 | 0.32 | 1.44 | 3.56 | 0.073 |
| 7. Women (%) | 3.11 | 3.91 | 2.52 | 0.50 | 0.50 | 3.11 | 1 | 2.08 | 4.64 | 0.162 |
| 8. Literacy rate (%) | 0.79 | 2.62 | 0.79 | 0.28 | 0.30 | 0.69 | 0.48 | 1 | 3.00 | 0.068 |
| (Consistency Ratio=0.024, Radom Inconsistency=1.41) | | | | | | | | | | |

**Table 11:** Aggregated Pairwise comparison matrix, weight and consistency ratio of composite earthquake vulnerability parameters based on the expert's opinion

| Composite index | Geo-logical | Structural | Systematic | Socio-economic | Weight |
|---|---|---|---|---|---|
| Geo-logical | 1 | 2.29 | 2.29 | 3.92 | 0.459 |
| Structural | 0.45 | 1 | 1 | 2.62 | 0.223 |
| Systematic | 0.45 | 1 | 1 | 2.62 | 0.223 |
| Socio-economic | 0.26 | 0.38 | 0.38 | 1 | 0.095 |
| (Consistency Ratio=0.01, Random inconsistency =0.9) | | | | | |

In this study 24 vulnerability parameters are weighted on a scale of 0 to 1. It is essential to assign a weight to every sub-category of the abovementioned 24 parameters. Providing different weight to every sub-factor is a complex task and time consuming also. This study classifies each of the vulnerability parameters into three categories viz., low, medium and highly vulnerable. Based on the recommendation of the experts and literature review (Islam, Swapan, and Haque, 2013), the subcategories are weighted in a scale of 0 to 1 where the weight of highly vulnerable category is 0.500, the medium vulnerable category is 0.333, and the low vulnerable category is 0. 167. The framework used for earthquake vulnerability assessment of Mymensingh city is shown in **Fig.2**.

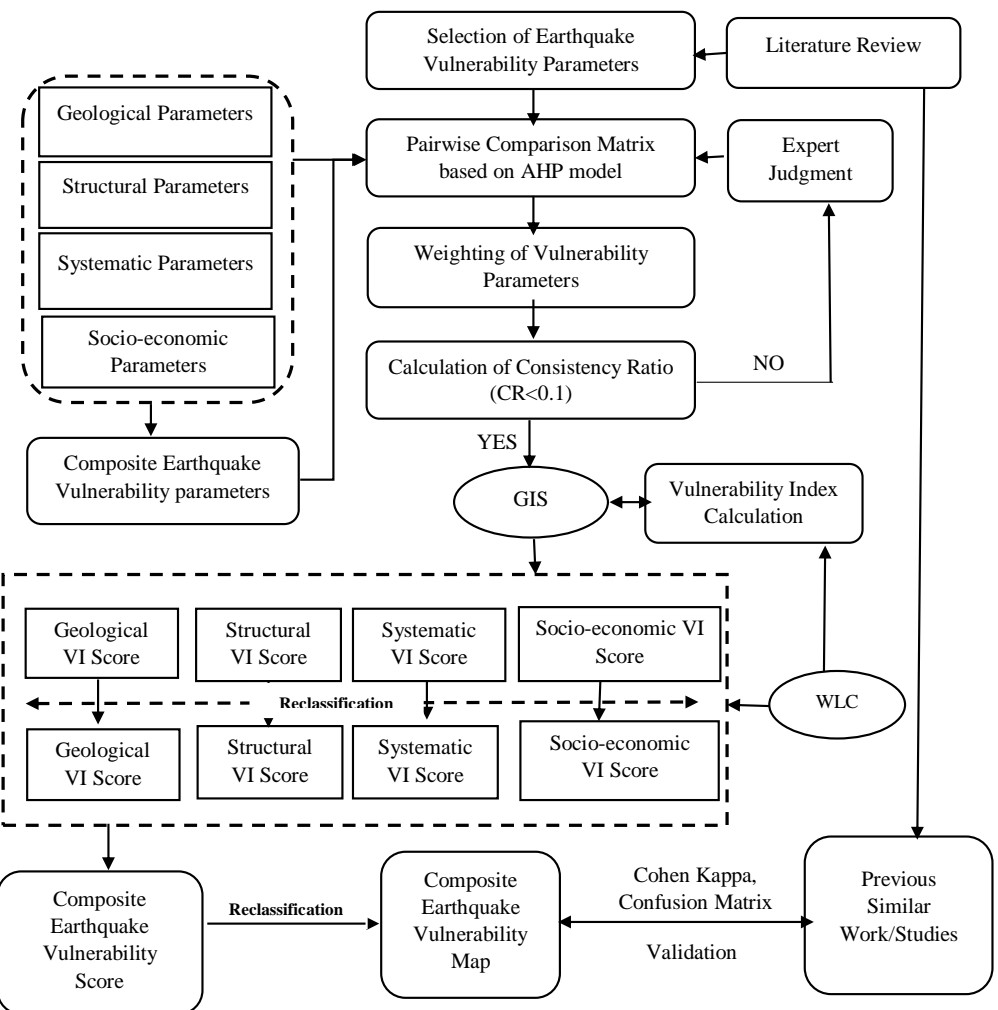

**Fig. 2:** Framework of composite earthquake vulnerability assessment

### 3.4. Data Source

In this study, Databases of Mymensingh Strategic Development Plan (MSDP), 2011-2031 prepared under Comprehensive Disaster Management Programme (CDMP)-II of the Ministry of Disaster Management and Relief and Urban Development Directorate (UDD), Ministry of Housing and Public Works, Bangladesh (UDD,2016) has been used. Data of structural parameters are collected from the physical feature database, land use database, and road network database of MSDP. Data of geological and socio-economic parameters are collected from the geological and socio-economic survey database of MSDP respectively. To calculate systematic vulnerability index, distances of each of the neighbourhoods from important facilities are calculated through employing a Network Analyst tool of proprietary ArcGIS, using point feature database of MSDP.

### 3.5. Data Analysis and Vulnerability Maps Preparation

In this study, the Analytical Hierarchical Process has determined weights of different factors and sub-factors of seismic vulnerability. All gathered data has been processed in the following sequential order: Firstly, the socio-economic data and vulnerability scores of earthquake vulnerability of Mymensingh city has been stored in SPSS environment and converted into Microsoft Access database to make them usable for analysis in GIS software

(ESRI product ArcGIS has been used). Secondly, neighbourhood wise data of structural and geological
earthquake vulnerability of Mymensingh city have been extracted using geo-processing in the ArcGIS
environment. Then, the databases are joined with the residential neighbourhood map of Mymensingh city map
in vector-based GIS. The centre points of each residential neighbourhoods are delineated using the conversion
tool in ArcGIS. In the next step, the maps have been reproduced for determining systematic vulnerability
parameters using closest facility function under Network Analyst tool in proprietary GIS software to identify
neighbourhoods which are inaccessible or possess less accessibility to the hospital, fire service, emergency
shelter, and evacuation route. The score of systematic earthquake vulnerability is reclassified and joined with the
residential neighbourhood map of Mymensingh city in vector-based GIS. Finally, the composite earthquake
vulnerability map of the residential neighbourhoods of Mymensingh city is produced using WLC method based
on reclassified composite vulnerability score in the ArcGIS environment (**Fig.3**).

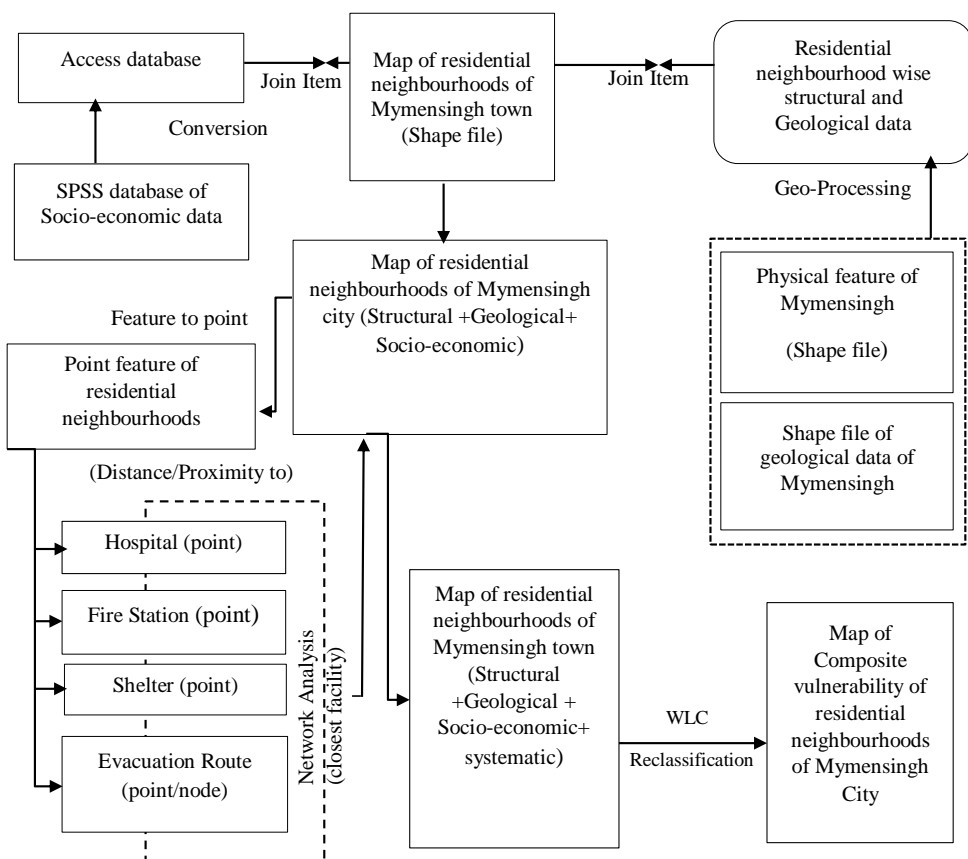

**Fig. 3:** Steps in GIS analysis

**3.6. Validation Methods Adopted**
Cohen kappa statistics and confusion matrix methods are used in this study to compare the result of this current
study with other similar studies.
The Cohen kappa statistic, well-recognised accuracy assessment algorithm mostly used to assess the
performance of the classifier, is a metric that compares an Observed Accuracy with an Expected Accuracy and
illustrates the agreement between two accuracy results on a scale of 0 to 1. Cohen kappa score 1 indicates
complete agreement and values 0 indicate no agreement between the two results. In this study, a comparison




between the result of other similar studies (observed accuracy) and the result of this study (expected accuracy)
are done using the Cohen kappa statistic. The vulnerability map of other similar studies and the composite
vulnerability map of the current research need to be converted into 1m× 1m raster grid to measure the agreement
using Cohen kappa. Cohen kappa statistics follow several steps. In the first step, a 2×2 metric is developed
based on the results, and observed accuracy ($P_o$) is determined by summing the total number of agreement and
dividing it by the number of total cells. In the second step, expected accuracy ($P_e$) is calculated by multiplying
the probability of agreement between high vulnerability cells of two similar studies with the probability of
agreement between low vulnerability cells. In the final step, the Cohen kappa score is calculated using the
following equation (6).
$$\text{Cohen Kappa} = \frac{Po - Pe}{1 - Pe} \tag{6}$$

Here, $P_0$ and $P_e$ represents observed accuracy and expected accuracy respectively. Pontius (2002) and Sousa et
al. (2002) suggested that kappa score less than 0.4 indicates poor performing models, 0.4 to 0.6 are fair,0.6 to
0.8 are good, and kappa score greater than 0.8 represent excellent agreements between expected model and
observed dataset.
Confusion matrix, also known as error matrix, is a spatial contingency table used to describe the performance of
a classification or prediction model on a test sample which true values are known and predicted or classified
sample. This table provides four different combinations of predicted and actual values. True Positive (TP)
indicates the prediction is positive and it's true whereas true negative (TN) means prediction is negative and its
true. On the contrary, false positive (FP) signifies the prediction is positive and its false whereas false negative
(FN) denotes prediction is negative and its false. Confusion matrix can be easily interpreted using **Fig. 4**.

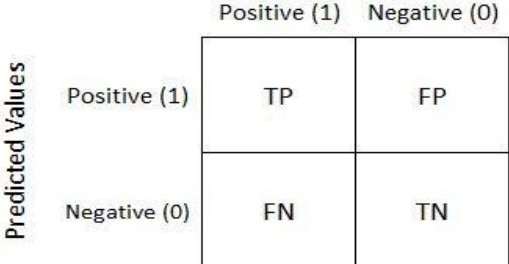

**Fig. 4:** Confusion Matrix classification system

## 4. Result and Discussion

The spatial variations of vulnerabilities are analyzed and shown in maps in 3 vulnerability zones, viz., high,
medium and low. From the city planning context for better understanding of the priorities of risk mitigation
activities, it is also essential to identify the relative importance of vulnerability parameters influencing
earthquake vulnerability of the neighbourhoods and therefore, have also been discussed in the following section
as well.

### 4.1. Geological Vulnerability

According to the geological dimensions, vulnerability analysis shows that 44 residential neighbourhoods are in
highly earthquake-vulnerable, 175 residential neighbourhoods are in medium earthquake-vulnerable; and only
22 neighbourhoods fall in low vulnerable zones in Mymensingh City. The spatial variation of geological
earthquake vulnerability of residential neighbourhoods of Mymensingh City is shown in **Fig.5**.

**Geological Earthquake Vulnerability Map of Mymensingh City**

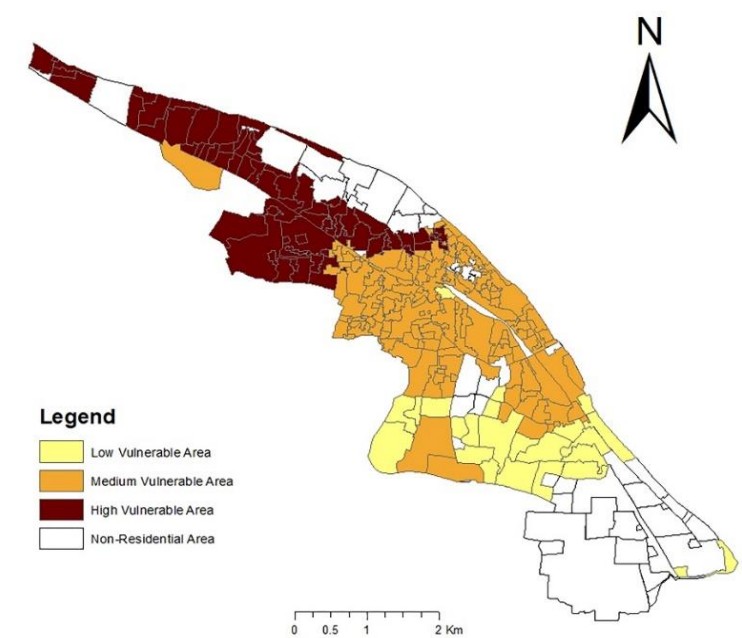

**Fig. 5:** Geological Vulnerability Map of Residential Neighbourhoods of Mymensingh City

**Fig. 6** shows the influences of different geological parameters on earthquake vulnerability (on a scale of 0-1). It
is observed that   Soil type has the highest (0.5) influence among the parameters followed by PGA (0.32).  Shear
Wave Velocity (0.18) has the least influence among the three parameters used in this analysis.

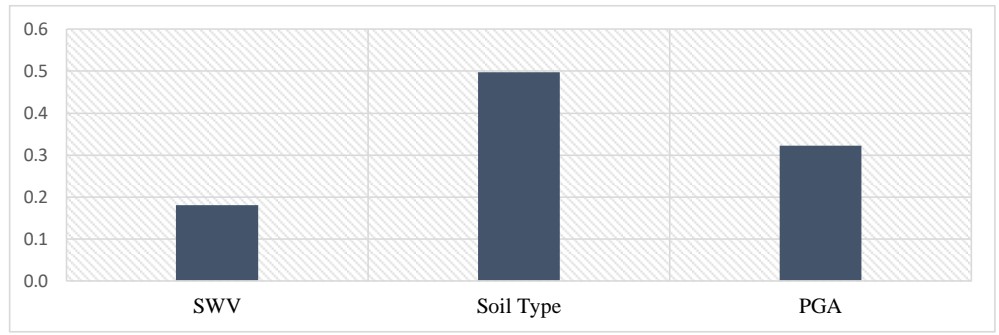

**Fig. 6:** Influence of Geological-Parameters on Earthquake vulnerability in Mymensingh city

**4.2. Systematic Vulnerability**
The distances of the hospital, fire station, emergency shelter and emergency evacuation route from the
geometric centre of each neighbourhood are considered and analysed in ArcGIS environment to assess the
spatial variation of systematic vulnerability. The result shows that 88 residential neighbourhoods of
Mymensingh city are stuated in the high earthquake-vulnerable zone as far as a systematic dimension of
earthquake vulnerability is concerned with feeble connections with these four emergency facilities. About 90
residential neighbourhoods of Mymensingh city fall in the medium systematic vulnerablezone.   Only 63


residential neighbourhoods, which have close spatial links with the above mentioned facilities, are in the low
systematically earthquake-vulnerable zone (**Fig. 7).**

**Systematic Earthquake Vulnerability Map of Mymensingh City**

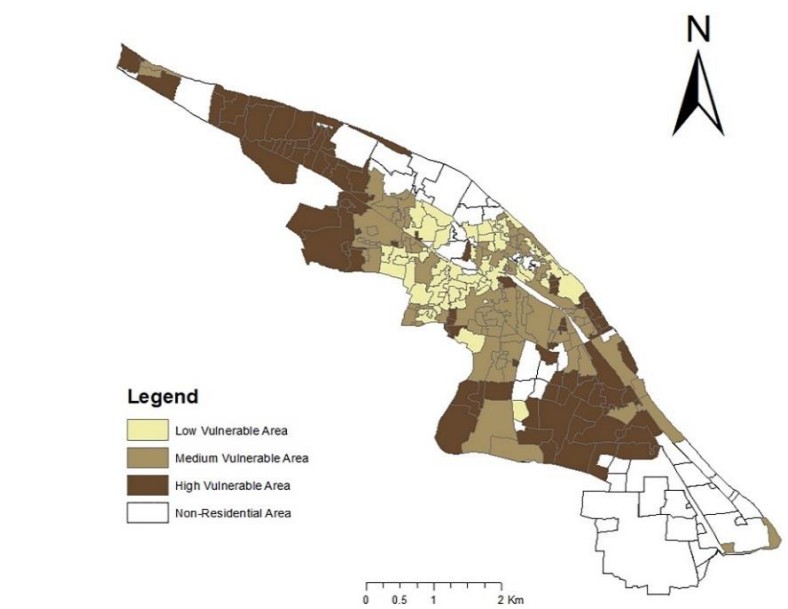

**Fig.7:** Systematic Vulnerability Map of Residential Neighbourhoods of Mymensingh City

The parameter wise assessment of systematic earthquake vulnerability of Mymensingh City on a scale of 0-1 is
shown in **Fig.8**. According to **Fig.8**, most of the residential neighbourhoods in Mymensingh City are highly
vulnerable due to their long distances from fire service stations (0.43), hospitals (0.24) and emergency shelter
(0.2) respectively.

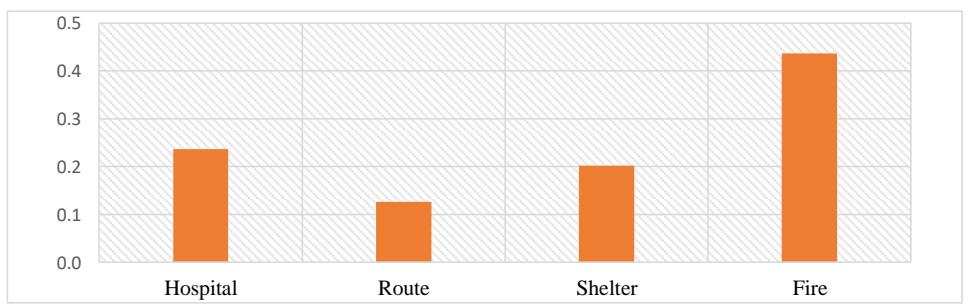

**Fig. 8:** Influence of Systematic Parameters on Earthquake vulnerability in Mymensingh city

**4.3. Structural Vulnerability**
From the analysis, it is found that eight residential neighbourhoods of Mymensingh city are highly structural
vulnerable, 54 residential neighbourhoods are medium structural vulnerable and 179 residential neighbourhoods
are low structural vulnerable.  It is interesting to observe that in Mymensingh city neighbourhoods, which are
structurally vulnerable, are not geologically vulnerable. The reason behind this difference is the location of the

**Structural Earthquake Vulnerability Map of Mymensingh City**

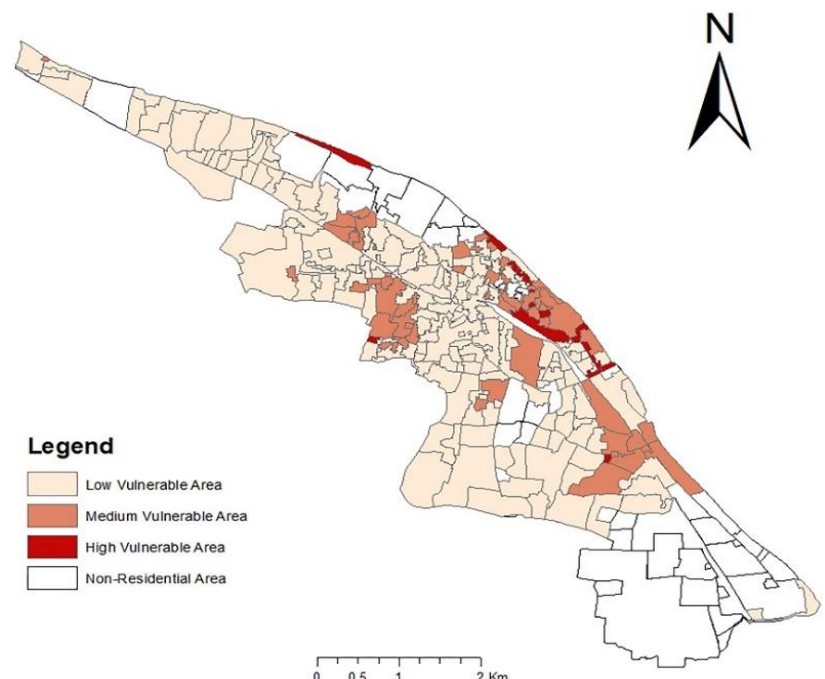

**Fig. 9:** Structural Earthquake Vulnerability Map of Mymensingh City

CBD area in the middle part of the city which is medium geologically vulnerable. In Mymensingh city, the
vulnerability parameters that make a city structurally vulnerable are comparatively high in the residential
neighbourhoods within or close to the CBD area than the neighbourhoods of other parts of the city. The spatial
variation of earthquake vulnerability of the residential neighbourhoods of Mymensingh city according to
structural dimension is shown in **Fig.9.**

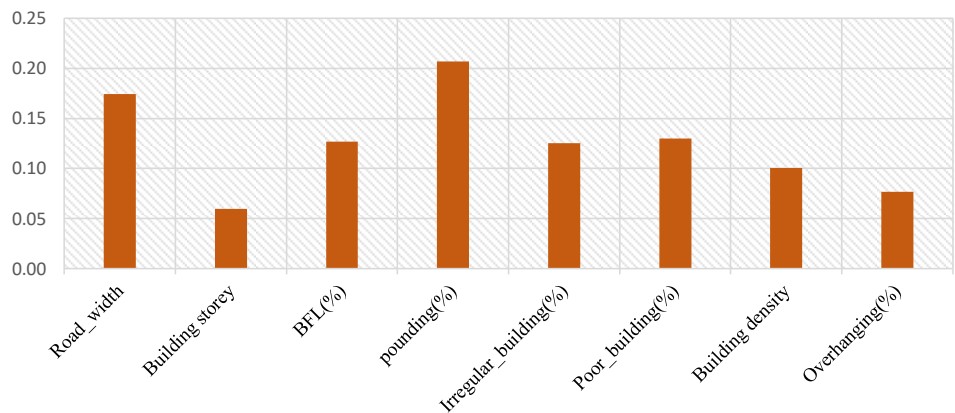

**Fig. 10:** Influence of Structural Parameters on Earthquake vulnerability in Mymensingh city





It is critical to know which parameter has the most influence onthe structural vulnerability to prioritise city
planning implications. **Fig.10** illustrates that the influence of 8 structural vulnerability parameters on overall
structural vulnerability (measured on a scale 0-1) and it is found that high pounding possibility (0.21), low road
width (0.17), a high percentage of poor building (0.13), irregular (0.13) and BFL buildings (0.13) respectively
are the primary reasons behind structural vulnerability in Mymensingh city.

### 4.4. Socio-economic Vulnerability

To get a complete picture of vulnerability situation of Mymensingh city, it is also essential to understand the
socio-economic characteristics of people living in different neighborhoods of the city. The result shows that 75
residential neighbourhoods of Mymensingh City are highly earthquake vulnerable from the socio-economic
context whereas 158 residential neighbourhoods are medium earthquake-vulnerable. Only eight residential
neighbourhoods are in a low vulnerable category in Mymensingh City. The spatial distributions of socio-
economic earthquake vulnerability in Mymensingh City are visually represented in **Fig. 11.**

**Socio-Economic Earthquake Vulnerability Map of Mymensingh City**

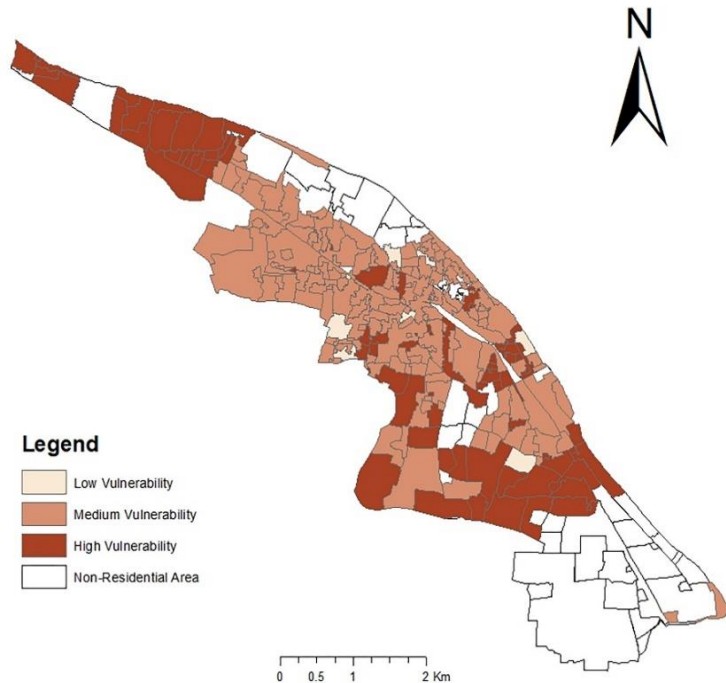

**Fig.11:** Socio-Economic Earthquake Vulnerability Map of Mymensingh city

The parameter wise socio-economic vulnerability analysis (**Fig.12**) of the residential neighbourhoods of
Mymensingh City shows that mainly the city is socio-economically earthquake-vulnerable due to the high
percentage of the elderly population (0.32), a high percentage of the child (0.24) and women population (0.16)
and population density (0.07). Other parameters' contribution to socio-economic vulnerability is less than 0.05.
As Mymensingh city is one of the oldest city and remarkable economic hub of the country since British colonial
period, the percentage of the elderly population, child and women are higher in the neighbourhoods of the city
than the national urban area average of Bangladesh (BBS,2010) which make its residential neighbourhoods
more socio-economically vulnerable.


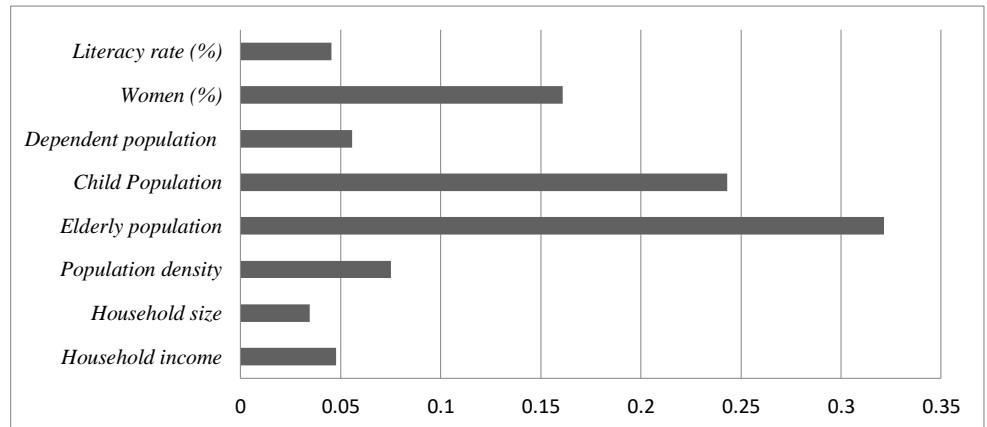

**Fig.12**: Influence of Socio-Economic parameters on Earthquake Vulnerability of Mymensingh City

### 4.5. Composite Earthquake Vulnerability


The result of composite earthquake vulnerability index shows that 51 residential neighbourhoods of
Mymensingh are highly earthquake-vulnerable from all four dimensions of vulnerability. About 123 residential
neighbourhoods are medium earthquake-vulnerable, and 67 residential neighbourhoods are in the low
earthquake-vulnerable category. Spatial distribution of composite vulnerability in residential neighbourhoods of
Mymensingh City is shown in **Fig.13**.

**Composite Earthquake Vulnerability Map of Mymensingh City**

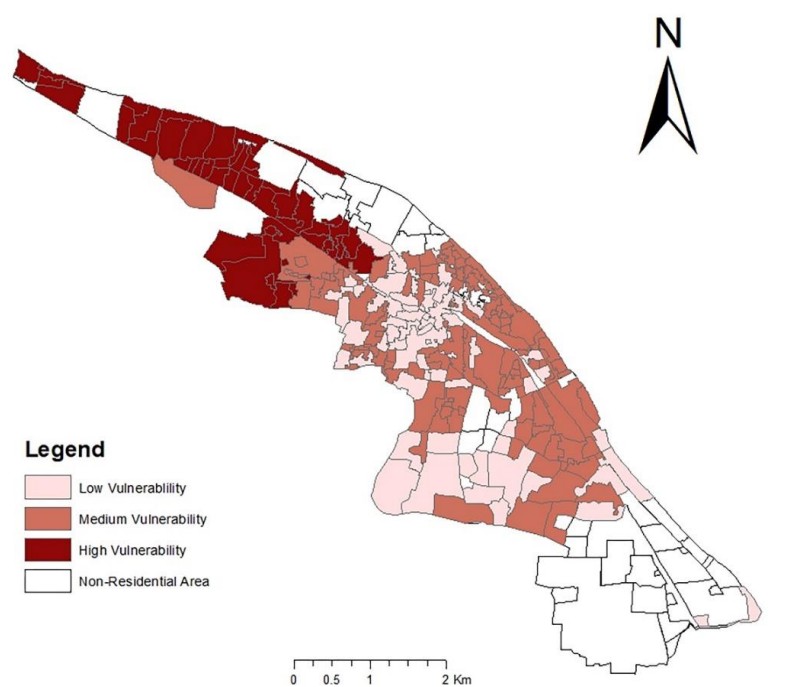

**Fig.13:** Composite earthquake vulnerability map of residential neighborhoods of Mymensingh city





In this study, 24 most important earthquake vulnerability parameters are considered to assess earthquake
vulnerability, and influence of each of the parameters on the composite earthquake vulnerability of Mymensingh
City are analysed and shown on a scale of 0-1. The concerned city planning and development agencies may
prioritise their earthquake risk reduction activities in Mymensingh City based on the influence of each of the
parameters on earthquake vulnerability as shown in **Fig.14**.

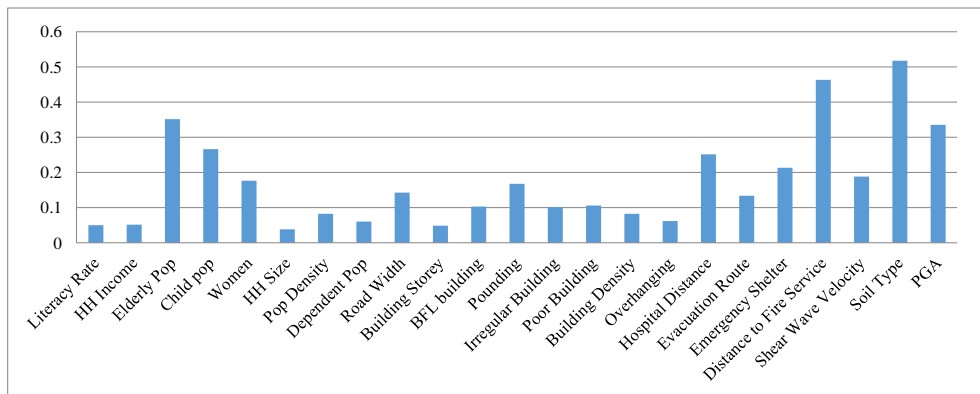

**Fig.14:** Influence of vulnerability parameters on composite earthquake vulnerability

According to the analysis, it is found that soil type (0.52), distance to the fire station (0.46), elderly population
(0.35), Peak Ground Acceleration (0.34), child population (0.27) and distance to hospital (0.25) respectively are
the topmost factors that make Mymensingh City highly earthquake-vulnerable. To be more specific, the
existense of 90% soft soil, only one fire station, high PGA value, a high percentage of elderly and child
population than national urban area average, spatial concentration of hospitals in the middle part of the city are
the main reason behind the earthquake vulnerability of Mymensingh city.
On the Contrary, household size (0.04), building storey (0.05), literacy rate(0.05), income per household (0.06)
and overhanging (0.06) has less influence on high earthquake vulnerability of Mymensingh city. Explicitly, high
percentage of muslim dominated neighbourhoods, small household size, high percentage of low rise buildings,
high literacy rate and income, etc. parameters are responsible for the low and medium earthquake vulnerability
of some residential neighbourhoods in Mymensingh.
**5.  Validation**
The composite vulnerability map, produced as an output of this research, has been compared with the output
similar other assements to observe the accuracy of the adopted methodology and to validate the applied method.
Comprehensive Disaster Management Program, phase-II (CDMP-II,2014) developed earthquake sensitivity map
for Mymensingh city using HAZUS methodology during the preparation of Mymensingh Strategic
Development Plan (MSDP), considering among other parameters PGA, spectral acceleration, foundation
condition, soil type, amplification factor, high and low-rise structure sensitivity (Haque,2015). The earthquake
sensitivity map developed by CDMP-II for Mymensingh city is shown in **Fig.15** in which the earthquake
sensitivity of Mymensingh city is classified into two categories viz. 1[st] degree and 2[nd]-degree earthquake
sensitivity. According to CDMP-II, 1[st]-degree earthquake sensitivity explicates the areas with high earthquake
hazard risk, and 2[nd]-degree earthquake sensitivity indicates the areas with low earthquake hazard risk.


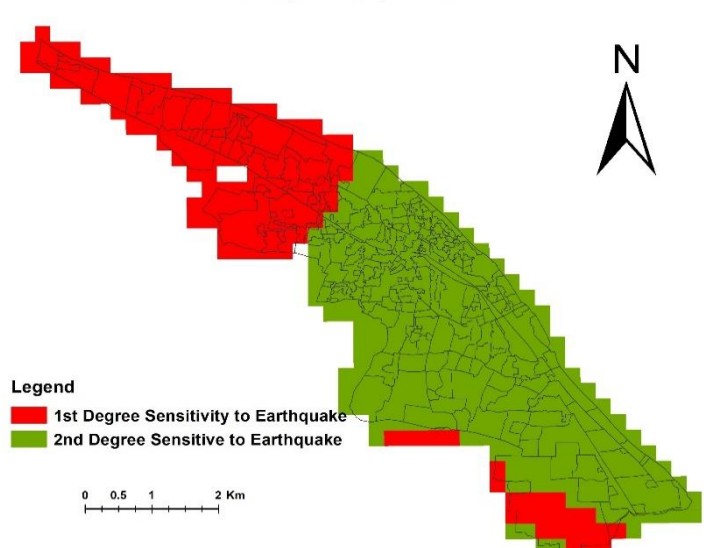

**Fig.15:** Earthquake sensitivity map developed by CDMP-II

Sarker, et al. (2009) did another work of earthquake risk assessment of Mymensingh city-based on SPT data of
boreholes, peak ground acceleration, site amplification, liquefaction and took the earthquake of 1897 as a
scenario event. In the seismic micro-zonation map of Mymensingh city, shown in **Fig.16,** high intensity
indicates high vulnerability. To compare the result of this study with results of CDMP-II, the result of this study
is classified into two categories viz. high earthquake vulnerability and low earthquake vulnerability where high
earthquake vulnerability represents the same highly vulnerable neighbourhoods and medium with low
vulnerable neighbourhoods jointly represent the low vulnerability. The result from CDMP-II (**Fig.15**) and
Sarker et al. (2009) (**Fig.16**) has been compared with the result of this study (**Fig.13**) using Cohen kappa
statistics and confusion matrix.
Applying equation (6), Cohen kappa score of this study, in comparison with CDMP-II is calculated, and the
score is found to be 0.6 which explicates that there is 60% agreement between the two results. According to the
kappa scale category, Cohen kappa score of this study falls in the good category which means there exist a good
agreement between the result from CDMP-II and the result of this study. Cohen kappa score of this study, in
comparison with Sarker et al. (2009) is found to be 0.53 indicating 53% agreement between two results and
which could be considered fair according to the scale of Pontius (2002).
The earthquake sensitivity map developed by CDMP-II mainly considered geology and infrastructure related
parameters and whereas in Sarker et al. (2009) only geological properties for seismic zonation was considered.
In both the studies very little attention has been given to the socio-economic context of the study area. On the
contrary, in the current study, vivid considerations have been given to the socio-economic dimensions of
vulnerability along with other dimensions which could be the main reason for disagreement of vulnerability
assessment among the mentioned results. The agreement and disagreement between high and low vulnerability
residential neighbourhoods of the two abovementioned results can be easily illustrated through the use of
confusion matrices.
Confusion matrix for CDMP-II map and vulnerability map of the current study is shown in **Fig.17**. Confusion
matrix without normalisation shows 2970 (60%) highly vulnerable cells of vulnerability map of the current
study are correctly classified and 1993 (40%) cells are falsely classified to low vulnerable zones which mean the
highly vulnerable area of this study has 60 percent similarity with CDMP-II produced vulnerability map.

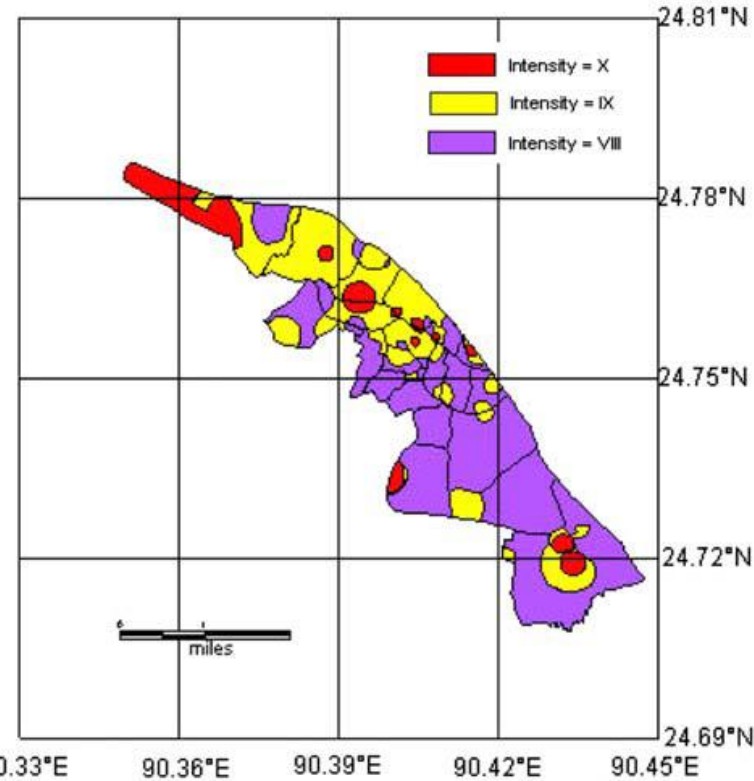

**Fig.16:** Seismic hazard intensity mapping of Mymensingh city (Source: Sarker et al., 2009)
Similarly, 10417 (94%) cells of low vulnerable zones of the current study are correctly classified in the low
vulnerability zone of CDMP-II map and 621 (6%) low vulnerability cells are falsely classified to the highly
vulnerable class of CDMP-II map which reveals that 94 percent of medium and low vulnerable area of this
study is similar to the 2nd-degree earthquake sensitive area marked by CDMP-II. The agreement or disagreement
between the result of this study and the result of Sarker et al. (2009) is also analysed using a confusion matrix.
The comparison of these two results is done only for residential cells. The confusion matrix score shows that
there exist 71% agreement in defining the highly vulnerable zones and 90% agreement in determining low

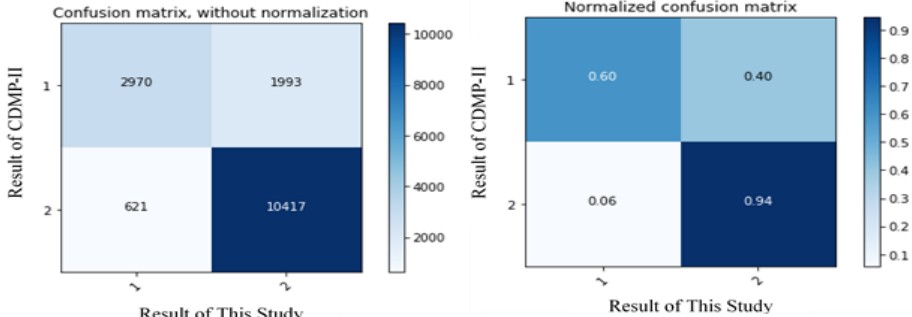

**Fig.17:** (a) Confusion matrix without normalization and (b) Normalized confusion matrix.
1=High Vulnerability and 2= Low Vulnerability




vulnerable zones (**Fig. 18**). The normalised confusion matrix shows that there exists 57% disagreement in
defining a medium vulnerable area which slightly misclassified as low vulnerable in the result.

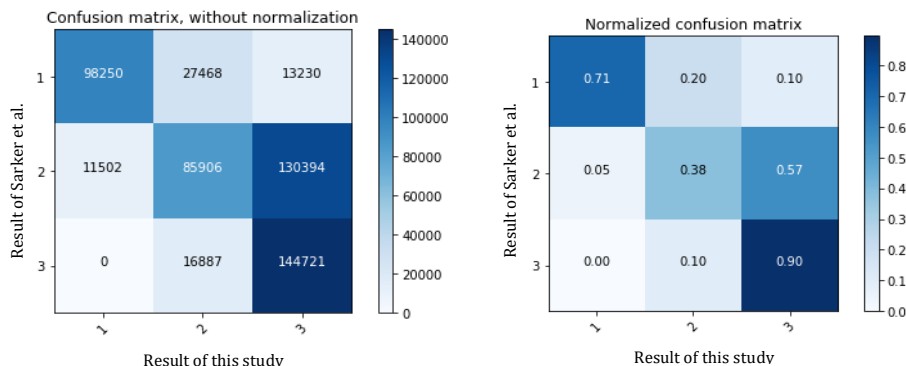

**Fig.18:** Confusion matrix (a) without normalization and (b) Normalized confusion matrix.
1=High Vulnerability,2= Medium Vulnerability and 3= low Vulnerability


## 6. Conclusion
Understanding spatial variability of earthquake vulnerability of a city in the earthquake susceptible zone is of
paramount importance for deciding on appropriate planning and development control interventions.
Incorporating earthquake risk in the city planning process for developing countries like Bangladesh is even more
challenging due to resource constraint, technological backwardness, deficiency of trained workforce, etc.
Though the HAZUS methodology is widely used for earthquake risk assessment, the methodology is found to
be of limited use in developing countries particularly in Bangladesh due to its enormous expertise, resource and
data support requirements. A more efficient, less resource and expertise consuming method needs to be
introduced for cities of developing nations which can assess earthquake risk with reasonable accuracy. This
paper introduced micro level land use specific earthquake vulnerability assessment methodology for
Mymensingh city with the application of GIS technology and employing an index-based approach which
follows several simple steps. The major strength of this method is its capability to provide a reasonably accurate
result of earthquake vulnerability and its spatial variation with minimum resource and expertise requirements.
The results by adopting the current AHP-GIS integrated approach is found to be reasonably accurate in
comparison with the results found by adopting the HAZUS methodology and the methodology suggested by
Sarker et al. (2009). Major advantages of using this suggested methodology for earthquake vulnerability
assessment are, it is cheaper, less time, resource and effort consuming and reasonably accurate for a city
planning application in the developing countries. This methodology can be applied in any earthquake-vulnerable
geographic location and expected to be helpful for policy makers in low-income countries to prioritise special
consideration area or hotspot for disaster management. The results of this paper are expected to be useful in
designing appropriate seismic risk reduction strategies for the local planning and development authorities.
**List of Abbreviations**
AHP= Analytical Hierarchy Process, GIS= Geographical Information System,WLC= Weighted Linear
Combination, FEMA= Federal Emergency Management Authority, CDMP= Comprehensive Disaster
Management Program, MSDP= Mymensingh Strategic Development Plan
**Declaration**
• **Availability of data and material:** The data used in this research is uploaded in a public
domain(http://www.msdp.gov.bd/ ) of government of peoples republic of Bangladesh
• **Competing Interest:** The authors declare that they have no competing interests
• **Funding:** This research got no funding from any sources.
• **Authors' contributions:** Md. Shaharier Alam analyzed and interprets the whole article with supervision of
Prof. Shamim Mahabubul Haque. All authors read and approved the final manuscript.



• **Acknowledgement:** Authors gratefully acknowledge assistances from Urban Development Directorate
(UDD)-Government of the People's Republic of Bangladesh and Comprehensive Disaster Management
Programme (CDMP) for data support for this research work. Authors also would like to thank Dr. Ishrat
Islam, Professor, Department of Urban and Regional Planning, Bangladesh University of Engineering and
Technology, Dhaka, Dr. Mehedi A. Ansary, Professor, Department of Civil Engineering, Bangladesh
University of Engineering and Technology, Dhaka and Naima Ahmed, Deputy Secretary, Ministry of Disaster
Management and relief, Dhaka, for their precious judgment as expert in this paper. Valuable suggestion and
comments from anonymous reviewers are also gratefully acknowledged.

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
