# Peer review of "Understanding Spatial Variations in Earthquake Vulnerabilities of"

_Natural Hazards and Earth System Sciences, 2019_

## Referee Comment (RC1) · Anonymous Referee #1 · 26 Nov 2019

I have read the invited MS very carefully. I believe that this study is report only and isn't suitable to publish this journal. Also, There are several papers with this subject in NHESS.

Introduction: - Line 37-43: You describe attribute of historical earthquakes. But you - should add pictures of earthquake, faults and etc. - Line 54-61: I don't find innovation this study!

[Figure]

Literature Review - This part is too long. This research isn't review search.

Methodology 3.1. Study Area - Fig. 1: please add coordinate. Also, position of city in Bangladesh country. - You should describe historical earthquakes by map and table. Also add depth of all earthquakes (it is very important).

3.2.1. Geological earthquake vulnerability parameters - calculating of PGA need several parameters such as length of fault, distant between fault and site, magnitude of earthquake, soil type, shear wave and etc. why do you use PGA, soil type, shear wave? There are many paper about calculating PGA and PSHA. For example:

Campbell, K. W. (2006). Campbell-Bozorgnia next generation attenuation (NGA) relations for PGA, PGV, and spectral acceleration: a progress report. Seism. Res. Lett., 77.‏

3.3.2. Weighted Linear Combination - Line 245-267: How do you determined rank of each parameter?

Method - How do you determined weight of each class of factor? I don't find anything about that.

- Line 305: There are several map with different scale. What do you utilize for convert scales to 1*1 m or 2*2 m? It is very important. - Fig 6, 8, 10, 12 shown influence of vulnerability parameters in each group. You calculate shown influence of vulnerability parameters by using AHP method. Why do you show again? Also, you shown that in Fig14. It is good because it is summery of all parameters.

Validation

- CDMP-II algorithm and Sarker algorithm aren't clear. It is too important for using and etc. - This study using several parameters such as social, urban and etc. data. But CDMP-II and Sarker algorithm utilized only geology data. How do you compare results?

Conclusion - This part is moderate. Consequently, I think that this part need add key important point about advantage and disadvantage AHP model and compare with HAZUZ completely.

---

## Referee Comment (RC2) · Anonymous Referee #2 · 10 Jan 2020

This manuscript proposes a method to study the spatial variability of the Earthquake Vulnerabilities in the city of Mymensingh using different parameters: geological, social, economic and structural. A hierarchical method (AHP) is used to assemble these 23 different indexes, by associating a weight to each parameter with the help of three experts. The results of this work are compared using a similarity method to the earthquake sensitivity map for Mymensingh developed by CDMP-II (2014) and and Sarker et al. (2009).

[Figure]

The aim of this work is to study the "Earthquake Vulnerabilities" but both in the abstract both in the paragraphs of paper it is not clear if it refers to the vulnerability, hazard or risk. The 23 indexes used in fact belong to different aspects of the risk. The authors choose to use some parameters and justify the non-use of others saying that they are not available in Mymensingh city. The method seems tailored to the database already used by authors (Alam & Haque, 2017) and not on the basis of how much represent the event studied.

The result of this work provide relative (not absolute) value of seismic vulnerability, and not quantify nor the number of people involved nor the possible fatalities caused by a seismic event; for these reasons it cannot be considered a risk assessment. At the same time the result cannot be defined a hazard assessment, since elements of structural vulnerability of buildings and social vulnerability are taken in account. The indexes with major weight are the geo-logical parameters, PGA and soil type, and that are referred to the earthquake hazard assessment.

The model proposed in this manuscript does not represent an advancement for the scientific community. First of all, the authors have to decide which aspect of the risk they want provide: hazard, vulnerability or risk. Secondly, the AHP and WLC methodology together with the use of GIS have also already been used in other works and the only novelty, also highlighted by the authors, is the use of low-cost data.

Some parameters belonging to vulnerability of buildings are "strange", in particular the height of the buildings is normally a parameter that indicates a better construction technique and therefore a better ability to resist a seismic event. Parameters as "Pounding", "Irregular Shape" and "Building and Heavy Overhanging" in Mynmensingh, where 87% of the buildings have only one floor (Alam & Haque, 2017) look trivial. As already mentioned, the choice of some parameters seems to be made only on the basis of the available database.

There are some text editing errors in the paper, for example: line 20, reseach; line 79,

counties; table 3, stroye.